# Thymoma-Related Paraneoplastic Syndrome Mimicking Reactive Arthritis

**DOI:** 10.3390/medicina57090932

**Published:** 2021-09-04

**Authors:** Chang-Hung Liao, Sin-Yi Lyu, Hsiang-Cheng Chen, Deh-Ming Chang, Chun-Chi Lu

**Affiliations:** 1Department of Internal Medicine, Tri-Service General Hospital, National Defense Medical Center, Taipei 114, Taiwan; joe50333@yahoo.com.tw; 2Division of Radiology, Tri-Service General Hospital Keelung Branch, National Defense Medical Center, Taipei 114, Taiwan; jusonlyu001@gmail.com; 3Division of Rheumatology/Immunology and Allergy, Department of Internal Medicine, Tri-Service General Hospital, National Defense Medical Center, Taipei 114, Taiwan; alex0624x@yahoo.com.tw (H.-C.C.); ming0503@ms3.hinet.net (D.-M.C.); 4Division of Allergy, Immunology, Rheumatology, Department of Internal Medicine, Taipei Veteran General Hospital, Taipei 114, Taiwan; 5Department of Pathology, University of Washington, Seattle, WA 98195, USA

**Keywords:** thymoma, paraneoplastic syndrome, reactive arthritis

## Abstract

*Background and Objectives*: Thymomas are associated with a high frequency of paraneoplastic manifestations. Paraneoplastic syndrome (PNS) with thymoma presents a challenge to clinicians because of the need to decipher the association between the presenting symptoms and the underlying tumor. The condition most commonly noted in patients with PNS with thymoma is myasthenia gravis. Other common autoimmune diseases that may present as PNS include systemic lupus erythematosus, pure red cell aplasia, and Good syndrome. Seventy-six percent of patients with PNS-associated thymoma experience resolution of PNS after curing thymoma. *Materials and Methods*: A 37-year-old man with a two-month fever accompanied by polyarthritis accidently found thymoma after contrast computed tomography scans of his chest. He accepted Video assisted thoracoscopic surgery with resection of thymoma. *Results*: Fever and polyarthritis resolved after operation but recurred in five days due to cytomegalovirus viremia, which might be predisposed by previous antibiotics treatment before the diagnosis of thymoma. *Conclusion*: Patients with a thymoma also have a high frequency of PNS, and the most frequent condition found in patients with PNS-associated thymoma is myasthenia gravis. Fever with polyarthritis has been rarely reported as a symptom of PNS-associated thymoma. Here we reported an unusual case of PNS mimicking reactive arthritis with thymoma, as diagnosed based on the patient’s clinical progression, imaging examination, and laboratory tests. The patient died of his comorbidities, and his death may have been related to long-term antibiotic use and consequent intestinal dysbiosis. This challenging case may help to inform clinicians of the need for detailed work-up of fever with unknown origin in the presence of chronic polyarthritis to prevent the overdiagnosis of inflammatory arthritis or rheumatic disease and avoid further comorbidities. Detailed work-up should include the patient’s history of infections, inflammation, and malignant or nonmalignant tumors.

## 1. Introduction

The clinical manifestation of paraneoplastic syndrome (PNS) associated with thymoma presents a challenge to clinicians because of the difficulty in deciphering the association between the presenting symptoms and the underlying mass, which is critical. A sizeable percentage (25–40%) of patients with thymoma present with myasthenia gravis, and more than 15% of patients diagnosed with thymoma present with a PNS presenting with fever and immunodeficiency like syndromes, such as Good syndrome, and hematologic disorders, such as pure red cell aplasia [1,2,3]. Simultaneous fever and polyarthritis can be diagnosed as inflammatory arthritis such as systemic lupus erythematosus (SLE), reactive arthritis or infectious arthritis [4], although neither of these syndromes is considered to be a manifestation of PNS with thymoma. Patients with PNS-related polyarthritis were reported in several solid tumors and hematologic malignancies except thymoma. Corticosteroids and disease-modifying anti-rheumatic drugs were typically used to treat this kind of patient but poor responses to treatment were observed [5]. The accurate diagnosis of patients with thymoma and PNS is a challenge. Here, we present a case involving a 37-year-old man who presented with simultaneous polyarthritis and fever of unknown origin lasting 2 months. Thymoma was diagnosed after a series of blood tests, imaging examinations, and pathology results.

## 2. Case Report

A 37-year-old man with a height of 1.64 m and weight of 70 kg had experienced intermittent spiking fever and chills for 2 months before this admission. He had no history of previous systemic disease and had lived overseas for 1 year before the first admission. He had no history of animal contact, unexplained wounds, or drug abuse and had no irritative symptoms of the airways, intestine, or brain. Polyarthritis had developed in all proximal interphalangeal joints of both hands and was accompanied by spiking fever and myalgia in both arms, chest wall, and back. Within 2 weeks after the onset of polyarthritis, these symptoms extended to the phalanxes of both feet, ankles, and knee joints. (Table 1).

His first admission was to the division of infectious diseases. The 2019 novel coronavirus disease (COVID-19) had been excluded after repeated blood sampling and plain X-ray films of the chest. Blood tests showed a white blood cell count of 12,850 cells/mm^3^, hemoglobin level of 13.3 g/dL, platelet count of 490,000 cells/mm^3^, plasma creatinine level of 1.0 mg/dL, alanine aminotransferase (ALT) level of 79 U/L, aspartate aminotransferase (AST) level of 38 U/L, and c-reactive protein (CRP) level of 15.00 mg/L. Serum tests did not identify viral hepatitis infection. Microbiology examination of the sputum, blood, urine, and feces did not identify any pathogens. Blood sampling did not detect any autoantibodies. Physical examination did not reveal abnormal murmurs of the heart or abnormal primitive neurologic reflexes. Plain X-ray films of the chest did not show mass lesions or abnormalities in the lungs. X-ray of the major joints of all limbs did not show fracture, erosion of bones, or abnormalities of soft tissue. A technetium-99m methylene diphosphonate bone scan revealed increased uptake in the left shoulder, elbow, wrist, and ankle, and right foot and toes, which was compatible with the clinically significant arthritis noted in the physical examination (Figure 1). The patient’s fever did not respond to antibiotics for 2 weeks from the first admission. Reactive arthritis was suspected initially because no significant pathogens or autoantibodies had been identified, and sterile inflammation was suspected. The fever responded to daily systemic methylprednisolone (40 mg), and the patient was discharged 17 days after the first admission.

However, polyarthritis in both wrists, metacarpophalangeal joints, knee joints, and ankles developed one week after the patient’s first discharge from the hospital. Physical examinations on the second admission revealed an alert consciousness without abnormal neurologic reflexes. A gallium-67 tumor scan indicated inflammation in the left shoulder, left sternoclavicular junction, left interscapular region, lateral right hip region, left buttock, right knee, and left ankle, but no evidence of malignancy (Figure 2).

The patient was admitted to the division of rheumatology for suspected partially treated reactive arthritis. Blood tests revealed a white blood cell count of 11,330 cells/mm^3^, hemoglobin level of 9.0 g/dL, platelet count of 639,000 cells/mm^3^, creatinine level of 0.6 mg/dL, ALT level of 20 U/L, AST level of 124 U/L, CRP level of 24.27 mg/L, ferritin level of 3408 ng/mL and erythrocyte sedimentation rate of 117/h. Again, microbial examinations of sputum, blood, urine, and feces did not identify any pathogens. Blood sampling did not detect any autoantibodies. Cardiac echo–Doppler revealed no vegetation at the cardiac valves. Sonography of whole abdomen revealed coarsening echo-pattern which was suspected to be fatty liver disease related. Based on the examination protocol for fever with unknown origin [6], computed tomography scans with contrast of the chest showed a mass lesion in the anterior mediastinum and enlarged lymph nodes (Figure 3).

The serum carcinoembryonic antigen concentration was 0.92 ng/mL (normal range: 0.00–5.00 ng/mL), squamous cell carcinoma antigen level was 1.40 ng/mL (normal range: 0.00–2.10 ng/mL), alpha-fetoprotein level was 0.83 ng/mL (normal range for males: 0.00–10.00 ng/mL), carbohydrate antigen 19-9 level was 4.22 U/mL (normal range: 0.00–37.00 U/mL), and beta 2-microglobulin level was 2.72 mg/L (normal range: 1.00–2.40 mg/L). Video-assisted thoracoscopic surgery (VATs) tumor resection identified a mass measuring 7.2 cm × 5.6 cm × 4 cm in the anterior mediastinum (Figure 4) and two adjacent lymph nodes.

Pathology analysis of the tumor revealed spindle-shaped epithelial cells mixed with small lymphocytes of thymic origin in the absence of malignant cells, and excisional lymph nodes were free of tumor metastasis. Benign thymoma, World Health Organization Type A, was diagnosed (Figure 5).

The patient’s spiking fever and polyarthritis subsided after the VATs tumor resection. The methylprednisolone dosage was tapered gradually from 40 mg/day to avoid adrenal insufficiency. However, the high fever returned 5 days after the operation and did not respond to antibiotics. Cytomegalovirus (CMV) viremia developed with a serum viral load of 5418 copies/mL, and ganciclovir was prescribed. Ileus and intermittent hematochezia developed gradually 8 days after the VATs (Figure 6).

CMV colitis was diagnosed based on a positive polymerase chain reaction test for CMV in the feces. A rapid test to detect Clostridium difficile revealed the presence of glutamate dehydrogenase antigen, and vancomycin and metronidazole were prescribed for the C. difficile infection. The patient developed fluctuating disturbed consciousness followed by intermittent seizure, and hematochezia developed 17 days after the tumor resection. Hyperammonemia (968 mg/dL) was noted, followed by status epilepsy. Blood tests showed a creatinine level of 0.5 mg/dL, ALT level of 36 U/L, AST level of 17 U/L, total bilirubin level of 0.9 mg/dL, direct bilirubin level of 0.3 mg/dL, and the absence of viral hepatitis infection. The patient received mechanical ventilation and emergency hemodialysis for metabolic encephalopathy. However, he developed a coma as a result of severe brain edema and died of subsequent brain death.

## 3. Discussion

Infectious arthritis, reactive arthritis after certain infections, and rheumatic diseases should receive a differential diagnosis when patients present with fever and polyarthritis. Reactive arthritis was suspected during this patient’s first admission. However, this patient did not have typical symptoms of infection of the urethra, conjunctiva, or intestine. Microbial examinations did not identify common pathogens of reactive arthritis such as Chlamydia trachomatis, Salmonella, Shigella, Campylobacter, or Yersinia species. The diagnosis of reactive arthritis also needs to exclude malignancy [7,8].

Patients with PNS-related polyarthritis are more likely to be male (male-to-female ratio of 7:1) and have a median age of 54 years. Several solid tumors and hematologic malignancies are associated with PNS-related polyarthritis. Lung cancer is the most frequent type of solid malignancy. In contrast to most patients with inflammatory arthritis, patients with PNS-related polyarthritis respond poorly to corticosteroids and disease-modifying antirheumatic drugs [5].

Thymoma is the most common lymphatic malignancy associated with PNS [3]. In the group of patients with PNS of thymoma, 63.1% of patients present with myasthenia gravis and 7.7% of patients present with pure red cell aplasia. Lichen planus (6.3%), good syndrome (5.9%) and lupus (4.1%) are also often presented of this kind of patient [1]. Cases involving thymoma with fever could be diagnosed as Good syndrome, for which invasive bacterial infection is a typical sign [9]. However, polyarthritis rarely occurs in patents with Good syndrome. Seventy-six percent of patients diagnosed with both PNS and thymoma before surgery experience remission of PNS after tumor resection, although thirty-three percent of these patients have recurrence of symptoms or onset of new PNS 1 month after tumor resection [1]. In our patient, fever and polyarthritis recurred 5 days after the VATs operation. In the literature, fever and polyarthritis mimicking reactive arthritis had never been reported in the context of PNS with thymoma. The patient had not responded to the treatment for reactive arthritis, including corticosteroid and sulfasalazine until VATs tumor resection [7].

CMV viremia and fever developed 5 days after VATs. CMV colitis, C. difficile-related pseudomembranous colitis and severe ileus were noticed 14 days after VATs. Hyperammonemia and status epilepsy developed 20 days after VATs tumor resection. No abnormalities were detected for the possible secondary causes of hyperammonemia include liver disease, renal disease, infection, and medication record. Congenital causes, such as inborn errors of metabolism [10], had been excluded because the patient had no congenital abnormalities or growth defects. CMV viremia and colitis, and pseudomembranous colitis were suspected as the major cause of this patient’s hyperammonemia. Long-term antibiotic and steroid use are risk factors for CMV and C. difficile colitis. Antibiotics had been prescribed for 2 months after the first admission because of the patient’s fever and suspected infections, especially because this patient was treated during the COVID-19 pandemic. We hypothesize that the risk of antibiotic-related CMV and C. difficile colitis may have been lower if the patient’s thymoma and associated PNS, which appeared as polyarthritis, had been identified earlier.

## 4. Conclusions

Thymoma is associated with a high incidence of PNS. Fever and polyarthritis mimicking reactive arthritis or rheumatic disease has seldom been reported in association with PNS with thymoma. This patient died of his comorbidities, which may have been related to his long-term antibiotic use. We report this rare, critical, and clinically challenging case to make clinicians aware of the need for the detailed work-up of patients presenting with fever with unknown origin. This will help clinicians provide an accurate diagnosis and prevent possible comorbidities and complications.

## Figures and Tables

**Figure 1 medicina-57-00932-f001:**
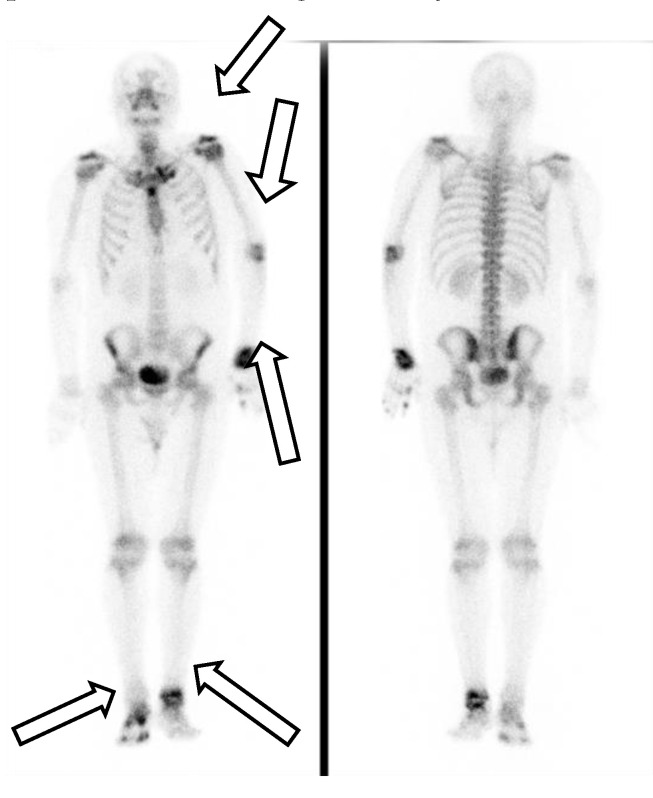
A technetium-99m methylene diphosphonate bone scan revealed increased uptake in the left shoulder, elbow, wrist, and ankle, and right foot and toes.

**Figure 2 medicina-57-00932-f002:**
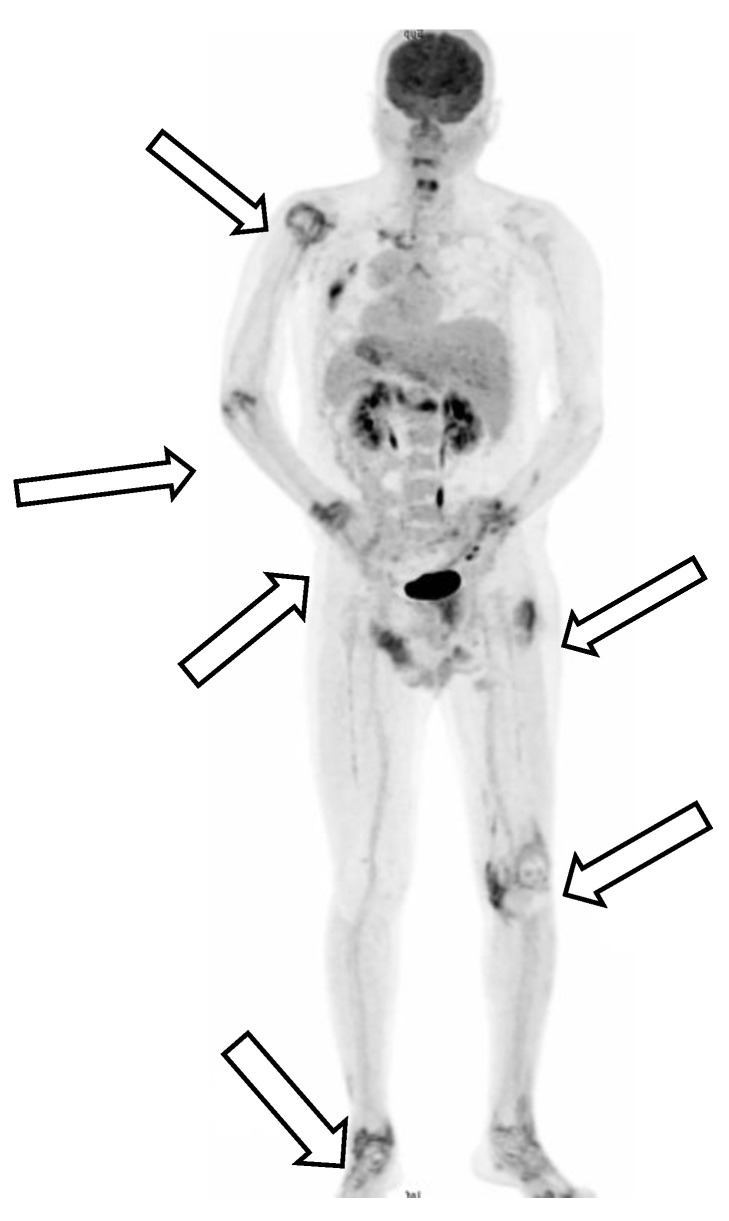
A gallium-67 tumor scan indicated inflammation over left shoulder, left sternoclavicular junction, left interscapular region, lateral right hip region, left buttock, right knee, and left ankle, but no evidence of malignancy.

**Figure 3 medicina-57-00932-f003:**
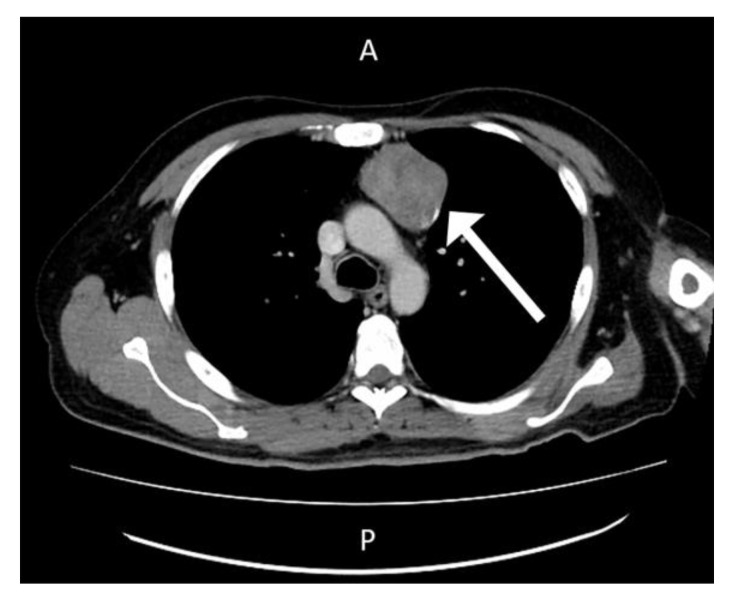
Computed Tomogram Scan with contrast at Chest revealed mass lesion over anterior mediastinum region.

**Figure 4 medicina-57-00932-f004:**
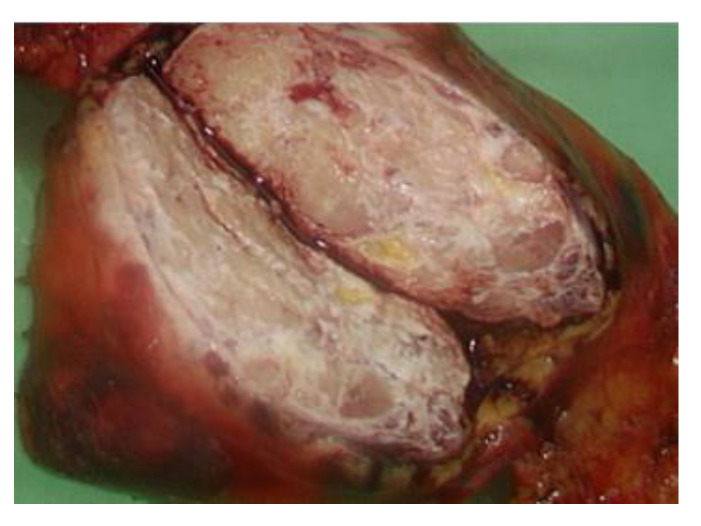
A 7.2 cm × 5.6 cm × 4 cm mass identified after operation of Video-Assisted Thoracoscopic Surgery.

**Figure 5 medicina-57-00932-f005:**
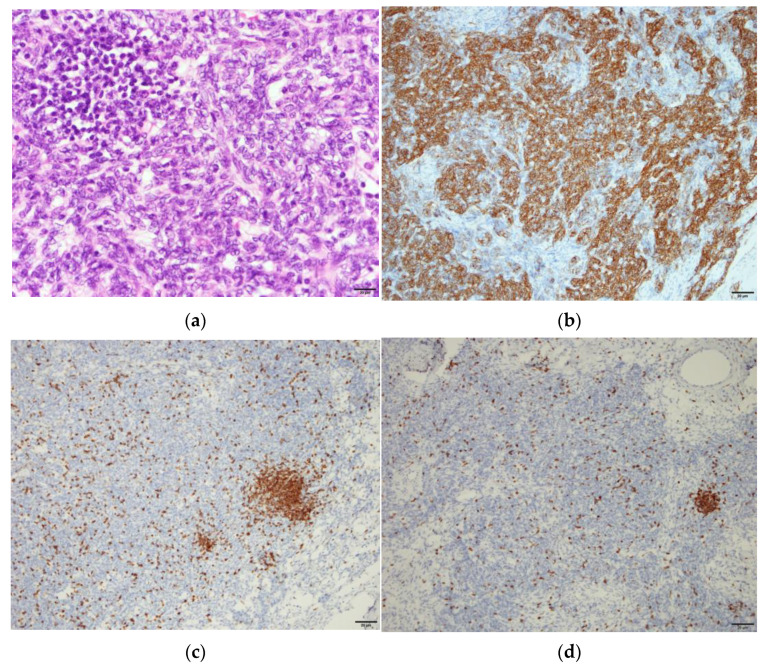
Microscopic appearance of the lesion. (**a**) Spindle epithelium mixed with small lymphocytes of the thymic tissue. (**b**) Immunohistochemistry (IHC) stains of cytokeratin revealed positive over epithelium component (**c**) IHC stains of CD 45 revealed positive for lymphocytes (**d**) IHC stains of CD5 revealed negative over epithelium component.

**Figure 6 medicina-57-00932-f006:**
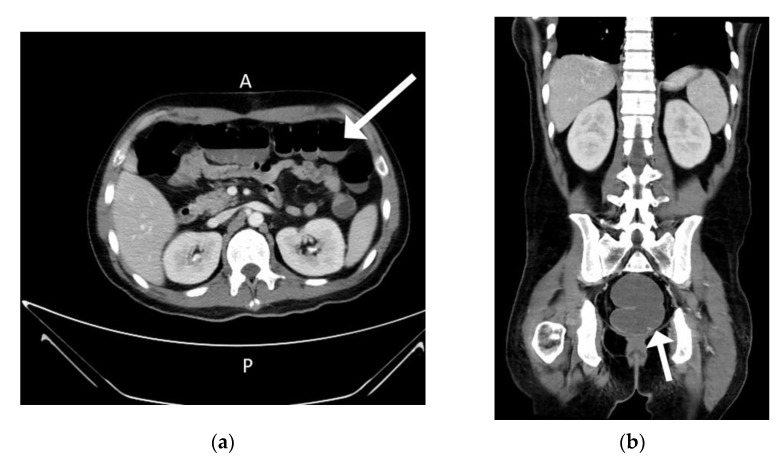
Ileus and hematochezia after 8 days of tumor resection. (**a**) Air-fluid level over transverse colon region (**b**) Sigmoid colon dilation.

**Table 1 medicina-57-00932-t001:** Health summary of patient before first admission.

Gender	Male
Age	37
Height	1.64 m
Body weight	70 kg
Past history	No systemic disease before
Personal history	No history of using behavior of Tobacco, Alcohol, and Betel Nuts
Travel history	Lived overseas(Australia) for 1 year before first admission
Occupation history	None
Contact history	None
Cluster history	None
Present illness	**Two months before first admission-** Intermittent spiking fever and chills **Two weeks before admission-** 1.Polyarthritis over proximal interphalangeal joints of both hands 2.Spiking fever 3.Myalgia in both arms, chest wall, and back **At first admission-** Above symptoms persisted and range of polyarthritis extended to the phalanxes of both feet, ankles, and knee joints

## Data Availability

The data that support the findings of this study are available from the corresponding author, C.-C.L., upon reasonable request.

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
