# Peer review of "Thymoma-Related Paraneoplastic Syndrome Mimicking Reactive Arthritis"

_medicina, 2021, doi:10.3390/medicina57090932_

Round 1

Reviewer 1 Report

The authors can rewrite the text to be more specific in their claims. 

  1. What percentage of patients have the co-morbidities listed by the authors? For example, what % of patients with PNS have lupus? or Myasthenia gravis? 
  2. Introduction needs to be expanded with other details from history. Has something similar been observed before? How are conditions like this commonly treated? 
  3. Initial health summary of the patient can be presented in a table
  4. For all images and scans, use arrows to indicate defect that authors talk about.

Author Response

Dear Reviewer:

Thank you for your insightful and constructive review of our article (medicina-1353027) titled “Thymoma related paraneoplastic syndrome mimicking reactive arthritis ". We appreciate the helpful comments provided during the course of review and we hope we have responded in a manner that satisfies each of the concerns that have been raised.

We have provided a point-by-point response to the comments of both reviewers below. We appreciate this opportunity to improve our manuscript and have endeavoured to craft a report of our findings that merits publication in Medicina.

Sincerely,

Chun-Chi Lu, MD, PhD 

Department of Internal Medicine

Tri-Service General Hospital

National Defense Medical Center

Taipei, Taiwan

Point 1: What percentage of patients have the co-morbidities listed by the authors? For example, what % of patients with PNS have lupus? or Myasthenia gravis?

Response 1: We thank the reviewer’s helpful comments. We added detailed information about percentage over each PNS of thymoma   In the group of patients with PNS of thymoma, 63.1% of patients present with myasthenia gravis and 7.7% of patients present Pure red cell aplasia. Lichen planus (6.3%), good syndrome (5.9%) and lupus (4.1%) are also major presentation of this patient.

Point 2: Introduction needs to be expanded with other details from history. Has something similar been observed before? How are conditions like this commonly treated?

Response 2: The reviewer raises a very important point. Patients with PNS-related polyarthritis were reported in several solid tumors and hematologic malignancies except thymoma. Corticosteroids and disease-modifying anti-rheumatic drugs were usually used to treat this kind of patients but poorly response of treatment were observed.

Point 3: Initial health summary of the patient can be presented in a table

Response 3: We thank the reviewer’s helpful comments. We had added a table entitled「Basic characteristic of patient」in our article to summarize patients initial health summary.

Point 4: For all images and scans, use arrows to indicate defect that authors talk about.

Response 4: The reviewer raises another excellent point. We had added arrow in Figure 1 and Figure 2 to indicate the defect of scans.

Reviewer 2 Report

Dear Authors

The case report "Thymoma related paraneoplastic syndrome mimicking reactive arthritis" is very interesting.

I have some questions:

  1. Did the authors check lipids, proteins, bilirubin, BHCG,  in serum?
  2. What was the cause for hyperammonemia in this case? Could the Authors explain it?
  3. What kind of treatment did the Authors use?
  4. It is necessary to include differential diagnosis of this case report.

Author Response

Response to Reviewer 2 Comments

Dear Reviewer:

Thank you for your insightful and constructive review of our article (medicina-1353027) titled “Thymoma related paraneoplastic syndrome mimicking reactive arthritis ". We appreciate the helpful comments provided during the course of review and we hope we have responded in a manner that satisfies each of the concerns that have been raised.

We have provided a point-by-point response to the comments of both reviewers below. We appreciate this opportunity to improve our manuscript and have endeavoured to craft a report of our findings that merits publication in Medicina.

Sincerely,

Chun-Chi Lu, MD, PhD 

Department of Internal Medicine

Tri-Service General Hospital

National Defense Medical Center

Taipei, Taiwan

Point 1: Did the authors check lipids, proteins, bilirubin, BHCG, in serum?

Response 1: We appreciate the reviewer’s opinion.  At first admission at INF ward, lipid profile revealed high-density lipoprotein(HDL) of 18 mg/dl, low-density lipoprotein(LDL) of 84 mg/dl and Triglyceride of 135 mg/dl. Albumin level revealed 3.5 g/dL at first admission and declined to 3.0 g/dL at second admission. Total bilirubin level revealed 0.6 mg/dL and Direct bilirubin level revealed 0.2 mg/dl at first admission. We had not checked BHCG during hospital course for this patient.

Point 2: What was the cause for hyperammonemia in this case? Could the Authors explain it?

Response 2: The reviewer raises a very important point.

We discussed the possible cause of hyperammonemia and listed the possible cause in our discussion part based on reference 10 in our manuscript. Several possible causes (including liver disease, renal disease, infection, medication record and congenital causes) had been ruled out for this patient. CMV colitis and pseudomembranous colitis were suspected as the major cause of this patient’s hyperammonemia.

Point 3: What kind of treatment did the Authors use?

Response 3: The reviewer raises another excellent point.  

Hemodialysis treatment had been operated for this patient for hyperammonemia. However, brain edema had been developed although hyperammonemia resolved after hemodialysis treatment.  It also reminded us to operate hemodialysis immediately when patient suffering from  hyperammonemia to prevent further co-morbidity.

Point 4: It is necessary to include differential diagnosis of this case report.

Response 4: We thank the reviewer’s helpful comments. We had added possible differential diagnosis of this case report in discussion part. Infectious arthritis, reactive arthritis and rheumatic diseases are differential diagnosis when patients present with fever and polyarthritis.